# Transient Abnormal Myelopoeisis and Mosaic down Syndrome in a Phenotypically Normal Newborn

**DOI:** 10.3390/children7060052

**Published:** 2020-05-28

**Authors:** Zachary Prudowsky, HyoJeong Han, Alexandra Stevens

**Affiliations:** 1Department of Hematology-Oncology, Texas Children’s Hospital, Houston, TX 77030, USA; hxhan@texaschildrens.org (H.H.); amsteven@texaschildrens.org (A.S.); 2Department of Pediatrics, Baylor College of Medicine, Houston, TX 77030, USA

**Keywords:** TAM, transient myeloproliferative disease, trisomy 21, down syndrome, mosaic down syndrome, *GATA1*, acute megakaryocytic leukemia (AMKL)

## Abstract

Transient abnormal myelopoiesis (TAM) is a common and potentially fatal neonatal complication of newborn babies with Down syndrome (DS). Children born with mosaic DS are also at risk of developing TAM. However, due to their variable phenotypes, early identification of patients with mosaic DS may be difficult; thus, early diagnosis of TAM is just as challenging. In this report, we describe a case of a phenotypically normal newborn who presented with concerns for neonatal leukemia. The diagnosis of mosaic DS and TAM was confirmed with abnormal *GATA1* mutation testing, highlighting the importance of early *GATA1* mutation testing in newborn leukemia with high suspicion for TAM.

## 1. Introduction

Transient abnormal myelopoiesis (TAM) is a leukemic condition in newborns classically associated with Down syndrome (DS), due to mutations in the *GATA1* gene. TAM presents in newborns with DS and blasts on peripheral smear. Severe cases may also present with cytopenias, hepatosplenomegaly, liver failure, coagulopathy, ascites, and pericardial effusion. Although the vast majority of cases self-resolve within the first 3 months of life, many patients require supportive care and even chemotherapy to survive the organ dysfunction associated with their initial presentation. Approximately 15%–23% of patients die from TAM due to the secondary liver or multiorgan failure [1,2]. Of those patients who survive TAM, approximately 20% will subsequently develop acute megakaryocytic leukemia (AMKL) within the first 4 years of life. Newborns with mosaic DS who develop TAM also have the subsequent risk of developing AML [3]. Children with mosaic DS may not have any distinguishing phenotypic features of DS, highlighting a diagnostic dilemma when a phenotypically normal newborn presents with findings which are concerning for leukemia [4]. Here, we present a case of a phenotypically normal newborn ultimately diagnosed with TAM and mosaic DS through the rapid implementation of a sequence of diagnostic tests with final confirmation on *GATA1* testing. This case highlights the benefit of including early *GATA1* testing in suspicious cases to promote the best clinical management and family counseling.

## 2. Case Presentation

This case follows a newborn female initially born as a monochorionic/diamniotic twin to a G3P3 mother by repeat C-section at 37 weeks and 2 days gestation. She and her twin were healthy after their delivery and were discharged after 2 days of hospitalization without complications, and they both received normal routine newborn care thereafter. However, on day-of-life (DOL) 8, the patient developed lethargy and increased work of breathing, prompting the parents to bring her to a local emergency room. There, she was found to be afebrile, but she was hypoxic to near 80% on room air, requiring a nasal cannula to regain normal oxygen saturation. Despite the resolution of her hypoxia, grunting, tachypnea, and lethargy persisted, and, additionally, jaundice was noted on the exam. The patient was admitted to the neonatal intensive care unit (NICU) for further workup.

Once admitted, blood cultures, urine cultures, and labs including a basic metabolic panel, liver panel, and complete blood count (CBC) were drawn. She was started on ampicillin, gentamicin, and acyclovir empirically. Labs were significant for the following: sodium of 130 (ref 137–145 mmol/L), unconjugated bilirubin 12.7 (ref 0.6–10.5 mg/dL), white blood cell count (WBC) 62.2 × 10^3^/μL (ref 5.0–38.0 × 10^3^/μL), and normal hemoglobin and platelet count. The manual differential of the CBC revealed 23% blasts, and subsequent repeat daily CBCs revealed a persistently elevated WBC with peripheral blast counts of 16%–21%. Flow cytometry from the peripheral blood revealed 47.5% CD34+ blasts with ambiguous lineage. On DOL 11 she was transferred to our tertiary care center for subspecialty management of suspected leukemia.

Once transferred to the tertiary care center NICU, a chest X-ray showed new cardiomegaly. A large circumferential pericardial effusion with tamponade was found on the echocardiogram, for which she underwent urgent pericardiocentesis and drain placement. A new diffuse erythematous papular rash was noted as well. Serum herpes simplex virus polymerase chain reaction and serologies were negative. No hepatosplenomegaly was found on an abdominal ultrasound. A CBC at this point showed a WBC 41.9 × 10^3^/μL (ref 9.1–34 × 10^3^/μL) with 33% blasts on manual differential, confirmed with a peripheral smear. Repeat flow cytometry of her peripheral blood showed 33% aberrant blasts which were positive for CD33, CD11b, CD34, CD117, CD99, CD58, CD38, and CD71, and negative for CD15, CD64, CD14, HLA-DR, CD52, CD25, myeloperoxidase, TdT, and cytoplasmic CD79a, concerning for megakaryoblastic lineage. Flow cytometry of the pericardial fluid only showed peripheral blood elements with eosinophilia and rare blasts. Liver function was normal aside from hypoalbuminemia.

With peripheral blood flow demonstrating blasts from the M7 linage, the possibility of TAM in a child with mosaic DS was considered. As the patient was critically ill, emergent chemotherapy was planned while the diagnostic workup progressed in parallel. Bone marrow analysis showed 20% marrow involvement of blasts with a high nuclear: cytoplasmic ratio, irregular contours, dispersed chromatin, and occasional cytoplasmic blebs and granules. The institutional fluorescence in-situ hybridization (FISH) panel of leukemia-related abnormalities was expedited. This panel included markers for *RUNX1* fusions. Although no fusions were seen, three copies of *RUNX1*, a gene located on chromosome 21, were noted. During this 24h period, the patient’s WBC and blast counts started to decrease with intravenous hydration alone; her rash and pericardial effusion improved and the drain was able to be removed. With this clinical improvement, and with the supportive evidence from the *RUNX1* FISH results, the likelihood of TAM rose on the differential despite the lack of DS features. To rapidly obtain additional data on the likelihood of trisomy 21 presence in the marrow, a centromere probe specific for the 21st chromosome was expedited, showing 23 of 100 nucleoli from the marrow with trisomy 21, a percentage that correlated with the degree of blast burden of the marrow at the time. Thus, a diagnosis of DS-associated TAM could still not be confirmed. 

Due to high suspicion of mosaic DS, *GATA1* sequencing was sent from peripheral blood at the same time as the centromere FISH testing. The child continued to improve without the initiation of chemotherapy. Repeat FISH and karyotype testing 5 days later when peripheral blasts were 12% by flow revealed normal karyotype of PHA-stimulated cells but 23% of cells with three copies of *RUNX1*. These findings again supported a possible diagnosis of TAM arising from a small trisomy 21 clone. One week later, the *GATA1* testing revealed a novel mosaic duplication within exon 2 from position 145–187, causing a frameshift mutation at codon 63 that introduced a premature stop codon at site 81. This particular mutation had not been described previously; however, other similar frameshift mutations have been well described in TAM and AMKL in DS, and the mutation was classified as being likely pathogenic [5]. Given this sequence of testing and the clinical features, TAM due to mosaic DS was definitively diagnosed. The child’s WBC and blast counts continued to decrease, her pericardial drain was quickly removed, and she was discharged to home on DOL 16. Two weeks after presentation, a repeat FISH and a karyotype on a peripheral blood specimen were done which were normal. Given her overall pattern of clinical findings, her *GATA1* mutation was presumed to be somatic given her lack of phenotypic features and resolution of her trisomy [6]. For this reason, additional mosaicism testing on cultured T lymphocytes and skin fibroblasts was not done. She has since been following up in the hospital’s outpatient cancer clinic for routine labs and screening. As of 3 years of age, she has not developed any signs of leukemia. Importantly, her twin also continues to do well with no signs of leukemia either. Both siblings met once with our genetics team as outpatients; however, no additional testing or genetic counseling has been recommended given the resolution of the genetic findings on the peripheral blood specimens and the lack of phenotypic findings on the exam.

## 3. Discussion

Non-DS neonatal leukemia, or leukemia presenting in a newborn less than 28 days of age, is a rare disease that typically accounts for <1% of all childhood leukemias and can be associated with several distinct somatic chromosomal abnormalities that drive leukemia, including *KMT2A* rearrangements [7]. Newborns with congenital leukemia typically require intensive multiagent chemotherapy to achieve the best chance of a cure. These agents are myelosuppressive and have the potential for significant organ toxicity. Only in very rare cases do congenital leukemias appear to self-resolve, and these cases do not often have any single unifying cytogenetic abnormality [8,9,10,11,12,13]. This is in contrast to DS-associated TAM, which occurs in approximately 10% of newborns with DS [1,2]. 

As its name suggests, TAM is transient and will self-resolve, typically within the first 3 months of life. TAM consists of characteristic blasts that appear morphologically similar to those of AMKL, and the diagnosis is confirmed by the presence of a *GATA1* frameshift mutation in addition to trisomy 21 on cytogenetic analysis. Outcomes of TAM can range from asymptomatic to fatal, with the symptoms of the disease spanning from subclinical to multiorgan failure due to the high leukemic burden and organ infiltration [1,2]. Death is most frequently due to liver fibrosis and failure, which can persist even after the peripheral blasts have cleared. The presence of a *GATA1* frameshift mutation renders the blasts of DS-associated TAM exquisitely sensitive to cytarabine. Therefore, in life-threatening cases of TAM, low-dose cytarabine can be used as monotherapy to reduce the number of circulating blasts in an attempt to recover organ function until the disease process self-resolves [2]. 

TAM can be easily identified in patients with classic DS; however, the diagnosis can be challenging in patients with mosaic DS, especially those without any phenotypic features or other complications of DS. In prior cases, mosaicism has often been confirmed using cultured skin fibroblasts and T-lymphocytes along with a cytogenetic analysis of blasts. Williams et al. reported three cases of TAM in patients with mosaic DS, only one of which was confirmed with *GATA1* testing [4]. The three others were diagnosed based on trisomy 21 found on cytogenetics, with mosaicism confirmed with cultured skin fibroblasts and T lymphocyte testing. Kawase et al. reported a case of TAM in a similar patient with phenotypically normal mosaic DS; however, *GATA1* testing from bone marrow was done on DOL 17, almost 2 weeks after the peripheral blasts disappeared. Testing at this time was negative, which was retrospectively attributed to low blast counts at the time of testing [14]. Richards et al. reported two cases of TAM in mosaic DS without phenotypic features; however, both were diagnosed based on a karyotype analysis of blasts and neither reported *GATA1* testing [15]. In our case, we were able to confirm the diagnosis of TAM using early cytogenetics and *GATA1* specific mutation testing from the peripheral blood. Prompt *GATA1* mutation testing not only confirmed the diagnosis but it also guided treatment away from using unnecessary intensive chemotherapy.

In 20–30% of cases, children who were diagnosed with TAM later developed DS-associated AMKL (AMKL-DS) within the first four years of life [1,2]. This risk is present for children with mosaic DS and TAM as well. Inaba et al. described a case where a patient with mosaic DS and TAM developed AMKL at 7 months of age [16]. For this reason, it is recommended to provide screening for these children for the first four years of life. In addition, the recommended treatment regimens and expected outcomes for children with AML vs. AMKL-DS are substantially different because of the cytarabine responsiveness of AMKL-DS blasts due to the presence of a driving *GATA1* mutation. Our patient still follows up in our outpatient oncology clinic, screening for AML with serial examinations and CBCs.

In summary, TAM as a feature of mosaic DS should be included in the differential for newborns presenting with signs and symptoms which are concerning for congenital leukemia, even without the typical features of DS. The presence of megakaryoblasts is a rare finding in congenital leukemia, but early awareness of its association with TAM can allow for the rapid initiation of diagnostic testing to distinguish the two diagnoses. Lastly, prompt *GATA1* testing of blasts before self-resolution can establish a definitive diagnosis, allowing for the effective counseling of families on both immediate and long-term management and screening.

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
