# Peer review of "Transient Abnormal Myelopoeisis and Mosaic down Syndrome in a Phenotypically Normal Newborn"

_children, 2020, doi:10.3390/children7060052_

Round 1

Reviewer 1 Report

The mosaic Down syndrome is proved by genetic testing of the baby's blood. Typically, 20 to 25 cells are examined. If some of the cells have trisomy 21 and some don't, then the diagnosis of mosaicism is made. However, this blood test can only determine the level of mosaicism in the blood cell line. While mosaicism can occur in just one cell line, it can also occur across other cell lines, such as skin cells. When mosaicism is suspected but not confirmed through the blood test, other cell types may be tested, skin and bone marrow are most commonly the next cells checked.  

Trisomy 21 is a frequently identified chromosomal aberration in human neoplasms, particularly in acute myeloid leukemia (AML) wherein it is the second most frequent trisomy after trisomy 8 (Strati 2013, Wagenblast 2019). Much of the literature regarding trisomy 21 in AML stems from the 10–20-fold increased risk of developing acute myeloid leukemia in patients with constitutional +21 or Down’s syndrome.

Since the patient and her sister did not have the phenotypic characteristics of Down's syndrome, and the percentage of cells with trisomy 21 was identical to AML blast, hence proving the true mosaicism of trisomy 21 is very important.

In 30% of newborns with Down syndrome, a transient pre-leukemia disease occurs, which is characterized by a clonal proliferation of immature megakaryocytes carrying somatic mutations in the GATA1 gene. Mutations in the activation domain of GATA1 are associated with TAM and acute megakaryoblastic leukemia of Down syndrome, while mutations in the N-ZnF motif of GATA1 and GATA1-S are associated with diseases similar to congenital dyserythropoietic anemia, congenital thrombocytopenia, and certain features that occur in thalassemia and myelofibrosis. In consequence inactivating mutation in the GATA1 gene cause diseases such as TAM, AMKL in Down Syndrome or Diamon-Blackfan anemia. Reduced levels of GATA1 due to reductions in the translation of GATA1mRNA into its transcription factor product are associated with promoting the progression of myelofibrosis.

The work is interesting. Nevertheless, taking into account the above-mentioned scientific evidence, I would ask the authors to answer for my questions and complete the following data beforehand:

  1. Full phenotype of leukemic cells,
  2. The result of the karyotype and all recommended FISH in the diagnosis of AML
  3. Indication of the exact type of GATA1 mutation in leukemic blasts. Is it somatic or germinal?
  4. Presentation of the actual analysis of mosaicism in healthy blood cells and/or epithelial/skin cells in the patient and her sister - I understand that the patient and the family have been undergoing genetic counselling and had these tests performed,
  5. Is the GATA1 mutation still present (congenital) in the patient and her sister? Have these tests been done?

Reviewer 2 Report

General comments:

This is a case report of a neonate without phenotype of Down syndrome (DS) who developed transient abnormal myelopoiesis (TAM). The authors argue that it is important to make a differential diagnosis of TAM by GATA1 gene mutation analysis in a non-DS newborn with leukemia, because one can avoid unnecessary intensive chemotherapy as mostly TAM resolves spontaneously. They point out an important issue, however, I think this has been already addressed in prior publications.

Minor comments:

  1. The authors should describe the morphological feature of the blasts and more detailed data of flow cytometry.
  2. Please provide the detailed data of the GATA1 gene mutation.
  3. Why did you examine RUNX1 by FISH prior to chromosome 21 using a centromere probe?
  4. I think it is necessary to examine FISH for chromosome 21in blood or skin fibroblast cells after remission of TAM to confirm the diagnosis of mosaic DS. What do you think about this point?

Round 2

Reviewer 2 Report

The authors responded and revised the manuscript properly according to the reviewers' comments. 

I missed to point out in the first review, but gene symbols should be italicized.
